# The Role of mRNA Alternative Splicing in Macrophages Infected with *Mycobacterium tuberculosis*: A Field Needing to Be Discovered

**DOI:** 10.3390/molecules29081798

**Published:** 2024-04-16

**Authors:** Weiling Hong, Hongxing Yang, Xiao Wang, Jingyi Shi, Jian Zhang, Jianping Xie

**Affiliations:** 1Jinhua Advanced Research Institute, Jinhua 321019, China; hongweiling1989@126.com (W.H.); hongxingyangbucm@163.com (H.Y.); 18042555826@163.com (X.W.); xmmlf@126.com (J.S.); 2Zhejiang University Medical Center, Hangzhou 311113, China; jian.zhang@zju.edu.cn; 3Institute of Modern Biopharmaceuticals, State Key Laboratory Breeding Base of Eco-Environment and Bio-Resource of the Three Gorges Area, School of Life Sciences, Southwest University, Beibei, Chongqing 400715, China

**Keywords:** *Mycobacterium tuberculosis*, alternative splicing, macrophages

## Abstract

*Mycobacterium tuberculosis* (*Mtb*) is one of the major causes of human death. In its battle with humans, *Mtb* has fully adapted to its host and developed ways to evade the immune system. At the same time, the human immune system has developed ways to respond to *Mtb*. The immune system responds to viral and bacterial infections through a variety of mechanisms, one of which is alternative splicing. In this study, we summarized the overall changes in alternative splicing of the transcriptome after macrophages were infected with *Mtb*. We found that after infection with *Mtb*, cells undergo changes, including (1) directly reducing the expression of splicing factors, which affects the regulation of gene expression, (2) altering the original function of proteins through splicing, which can involve gene truncation or changes in protein domains, and (3) expressing unique isoforms that may contribute to the identification and development of tuberculosis biomarkers. Moreover, alternative splicing regulation of immune-related genes, such as IL-4, IL-7, IL-7R, and IL-12R, may be an important factor affecting the activation or dormancy state of *Mtb*. These will help to fully understand the immune response to *Mtb* infection, which is crucial for the development of tuberculosis biomarkers and new drug targets.

## 1. Introduction 

Tuberculosis is an ancient infectious disease, the second most deadly infectious disease after COVID-19 [1]. Tuberculosis is caused by airborne transmission of the intracellular pathogen *Mtb*, which primarily infects macrophages in the alveoli [2]. The Global tuberculosis Report 2023 shows that 7.5 million people were diagnosed with tuberculosis in 2022, which is the highest record since the WHO began to monitor global tuberculosis in 1995 [3]. The global COVID-19 pandemic caused major setbacks in combating tuberculosis [4].

Nearly one third of the world’s population carries latent *Mtb*, but about one tenth will eventually develop into tuberculosis patients [5]. From this perspective, since the vast majority of people infected with *Mtb* do not develop the disease, there must be some powerful human anti-microbial mechanism preventing this. In the process of co-evolution, *Mtb* has been brilliantly adapted to the human host. It is clear that these pathogens, which cause chronic disease and persist in macrophages, must have acquired subtle strategies to resist host microbicidal mechanisms. Modulating the alternative splicing of host immune system-related proteins may be one of the effective strategies against host microbicidal mechanisms.

Alternative splicing in eukaryotic transcriptomes takes advantage of the use of different splicing sites and the use of differential exons/introns, resulting in transcription variations that produce different isoforms of specific proteins [6]. With the in-depth study of alternative splicing, researchers gradually realized that it has a regulatory effect on immunity. The host–microbial relationship affects the alternative splicing of immune-related proteins. In the study of tuberculosis, it has also been found that *Mtb* infection of macrophages causes great changes in the splicing of macrophages [7]. Alternative splicing of genes may affect whether infection occurs when exposed to *Mtb*, and whether infection can develop into a disease. Therefore, a comprehensive understanding of the alternative splicing of the human immune response to *Mtb* is essential for TB control and the development of new targets for vaccines and drugs.

## 2. Alternative Splicing

Constitutive splicing represents a crucial phase in RNA maturation, facilitating the excision of non-coding intronic sequences from the nascent transcript, thereby contributing to the generation of mature and functional messenger RNAs. To accomplish this essential task, the spliceosome—a complex ribonucleoprotein assembly consisting of various U spliceosomal RNAs and associated proteins—precisely identifies the splice junction and facilitates the execution of two crucial trans-esterification reactions, ultimately leading to the surgical removal of the intron from the pre-mRNA transcript. In a nutshell, the initial phase entails the specific interaction of U1 and U2 small nuclear ribonucleoproteins (snRNPs) with the 5′ splice site and the branch site, respectively, within the pre-mRNA transcript. During the subsequent step, the U4/U6-U5 tri-snRNP is recruited, and upon structural reorganization that follows, both U1 and U4 snRNPs are released, ultimately culminating in the formation of an activated B complex. The branch site approaches the 5′ splice site via the newly established connection between U6 and U2 snRNPs. Then, the 2′-hydroxyl group at the branch site initiates a nucleophilic attack on the 5′ phosphate of the first nucleotide of the intron, which constitutes the 5′ splice site and is typically a guanosine residue. The reaction cleaves the phosphodiester bond between the exon and the intron, while simultaneously forming a new phosphodiester bond between the branch site and the intron. During the second phase of the catalytic process, the recently liberated 3′ hydroxyl group initiates an attack on the ultimate nucleotide of the intron sequence, which usually consists of guanosine. This links the two exons together, resulting in the formation of the intron lariat (Figure 1a) [8,9,10,11].

In addition to the constitutive excision of introns, the mature transcript may also retain certain exons, exon fragments, or even introns through a formal process termed alternative splicing (AS), thereby complementing the constitutive intron removal. Gilber proposed alternative splicing (AS) of genes in 1978 [12]. His research has shown that the coding sequences of eukaryotic DNA are discontinuous and separated by non-coding sequences. These sequences are cleaved through different splicing methods to form mRNA with different sequences and functions. Early proposed a possible model for controlling the production of multiple transcripts by a single gene in 1982 [13]. Alternative splicing in plants was also discovered in 1989, and researchers speculated that this splicing method may be evolutionarily conservative in higher plants [14]. Alternative splicing is not only common in eukaryotes, but also exists in small amounts in archaea and bacteria [15,16]. Lower eukaryotes have fewer alternative splicing genes than higher eukaryotes, and vertebrates have higher alternative splicing genes than invertebrates [17,18]. Until the 1990s, scientists believed that only 5% of eukaryotic genes had alternative splicing [19]. With the development of the Human Genome Project (HGP) and Expressed Sequence Tag (EST), the possibility of analyzing alternative splicing of genomes based on bioinformatics has been realized. Pan used mRNA-Seq to analyze the complexity of alternative splicing in human tissues. New splicing sites were detected in about 20% of replicated exon genes, many of which were tissue-specific. Combined with EST-cDNA sequence data, they estimated that about 95% of human genes have alternative splicing [20,21].

Alternative splicing arises from a combination of stimulatory and inhibitory signals generated by multiple splicing factors in close proximity to weak splice sites. These signals either facilitate the assembly of the spliceosome at the specific location or destabilize it, resulting in a heterogeneous population of mature mRNAs. Factors that enhance splicing, including SR proteins, specifically associate with intronic or exonic splicing enhancers (ISE and ESE, respectively). In contrast, inhibitory factors like hnRNP proteins preferentially bind to intronic or exonic splicing silencers (ISS and ESS, respectively) (Figure 1b) [9,22]. The methods of alternative splicing mainly include exon skipping, alternative 5′SS selection, alternative 3′SS selection, mutually exclusive splicing, retained intron, alternative promoters, and alternative polyA [23,24,25,26]. Exon skipping is caused by the splicing of the exon and its two flanking introns from the transcript (Figure 2b), and is the most common alternative splicing method in mammals [27]. Exon skipping accounts for about 40% of the alternative splicing in higher eukaryotes, but is less common in lower eukaryotes [18,28]. Alternative 5′SS selection or alternative 3′SS selection occur when there are two or more splicing sites at the 5′ or 3′ end of the exon (Figure 2c,d). The proportion of alternative 3′SS selection and alternative 5′SS selection in higher eukaryotes is 18% and 8%, respectively [8]. Mutually exclusive splicing is the adjacent arrangement of two or more exons on the genome that contains only one of them when a mature mRNA molecule is formed by alternative splicing (Figure 2e). Mutually exclusive splicing is an important means of increasing the protein repertoire, by which the Down’s syndrome cell adhesion molecule1 (Dscam1) gene potentially generates 38,016 different isoforms. Our previous studies focused on the regulation mechanism and biological function of 38,016 isoform produced by mutually exclusive splicing of *Dscam1* [29]. Retained intron is co-spliced as an exon into the mature mRNA because the intron fails to splice in the pre-mRNA (Figure 2f). Retained intron often occurs in genes with short introns. Intron retention is common in plants, fungi, and protozoa [8,30], and is less common in vertebrates and invertebrates, with an occurrence less than 5% [6,28,31]. Alternative promoters are the phenomenon in which the first exon of a gene produces alternative splicing (Figure 2g). The alternative polyA refers to the selection of the alternative polyadenylate site of the last exon (Figure 2h).

## 3. Global Changes in Gene Expression upon Infection with *Mtb*

Alternative splicing can affect the stability, structure, function, and localization of transcripts, thereby altering the chemical and physiological characteristics of proteins. The transcriptome of macrophages during *Mtb* infection has been extensively studied. However, the response of alternative splicing in macrophage infection with *Mtb* has only been studied in recent years, but has not received enough attention. In fact, when macrophages are infected with *Mtb*, the alternative splicing in the transcriptome changes dramatically [7,32,33,34]. The alterations were global, with genes (see Appendix A) involved in functional categories such as splicing factor, immune response, autophagy, redox, and metabolism showing marked deviations in their splicing patterns in infected macrophages.

### 3.1. Alternative Splicing Causes Changes in Protein Activity

After macrophages are infected with Mycobacterium tuberculosis, the activity of some proteins is affected. This is because the splicing of these proteins has been altered, such as exon skipping, gene truncation mutations, and protein domain changes. The most significant splicing event observed after *Mtb* infection of lung epithelial cells was exon skipping, which was strain-specific [33]. Large amounts of truncated RAB8B variants contribute to *Mtb* survival by limiting RAB8B levels in cells [32]. The presence of premature stop codons hinders the translation of the truncated isoform, which in turn disrupts the autophagy process in macrophages and enables the virulent *Mtb* to evade elimination and persist. In addition, the expression of functional protein domains is affected by the alternative splicing of transcripts during *Mtb* infection. For example, large changes in the domains of most regulatory proteins in all kinases [7]. It is speculated that the kinase domain changes caused by the change of alternative splicing are mainly due to the important role of kinases in signal transduction. Exon skipping, truncated variants, and domain changes may cause proteins to lose their original activity. This suggests that *Mtb* may enhance its intracellular growth and reproduction through protein inactivation. We speculate that the regulation of cell alternative splicing during *Mtb* infection may serve as a target of anti-tuberculosis treatment.

### 3.2. TB-Specific Alternatively Spliced Isoforms

In addition, some genes exhibit tuberculosis-specific alternatively spliced isoforms, which have the potential to serve as biomarkers for the diagnosis of tuberculosis. The ABCC2, TNFAIP2, SLC26A11, IFI27, IFIT3, OAS1, OASL and GNB1 genes have a variety of TB-specific alternatively spliced isoforms, which may help explain the pathogenesis of TB and differential strain response [33]. Differential splicing of bacterial host-related TNFAIP2 transcripts may contribute to bacterial replication and persistence in lung epithelial cells [33]. S100A8-intron1-retention intron, RPS20-exon1 alternative promoter, KIF13B-exon4-skipping exit (SE), and UBE2B-exon7-SE as specific expressions can be used as molecular markers for the diagnosis of tuberculosis [35]. Existing diagnostic methods for tuberculosis often suffer from extended processing times and limited accuracy, making the exploration of promising biomarkers a consistent priority in tuberculosis research. Compared to traditional biomarkers for Mycobacterium tuberculosis infection, biomarkers developed through alternative splicing offer advantages such as higher diagnostic specificity [35], low cost, high efficiency, good reproducibility, and the possibility of early detection during infection.

### 3.3. The Expression of Splicing Factors Changes in Cells Infected with Mtb

The expression of splicing factors in cells infected with *Mtb* was inhibited. During *Mtb* infection, Protein-associated Splicing Factor (PSF) has been observed to be downregulated and plays a role in inhibiting apoptosis in *Mtb*-infected macrophages [36]. Serine/arginine-rich (SR) proteins play a pivotal role in the process of selective splicing and undergo tight regulation in diverse physiological as well as pathological settings [37]. After infection with H37Rv, the expression levels of SR proteins, SRSF2, SRSF3, SRp75, and SF3a in THP-1 cells were significantly downregulated [34]. The intricate interaction between viral and host SR proteins in regulating the expression of both viral and host genes has been extensively studied, revealing that viruses may efficiently exploit the host splicing machineries through the RS domain or RS-rich motifs present in their proteins, thereby benefiting their own gene expression [38]. In short, after infection with *Mtb*, the expression of splicing factors is downregulated, but the reasons for this change remain to be further studied.

## 4. Alternative Splicing in Host Response to *Mtb* Infection

### 4.1. The Role of IL-4/IL-4δ2 Ratio in Mtb Infection

IL-4, a multipotent cytokine, is one of the central regulatory factors of immunity in health and disease. It consists of four exons that encode the 15-kDa protein, composed of 129 amino acids. When the second exon is skipped by alternative splicing, it is described as IL-4δ2. IL-4δ2 has an antagonistic effect on IL-4 activity in human monocytes and B cells [39]. The mRNA level of IL-4 expression was higher in unstimulated peripheral blood mononuclear cells of tuberculosis patients [40]. High expression of IL-4 is thought to be detrimental to survival of *Mtb* infection [41]. The expression of IL-4δ2 was increased in latently infected healthy TB contacts (Figure 3) [41], but not in TB patients or uninfected individuals [42]. In whole blood from donors with pulmonary TB, the half-life of IL-4 mRNA was significantly prolonged by a factor of five, while IL-4δ2 mRNA was not (Figure 3) [43]. Progression from latent to active tuberculosis may depend on the potential ratio of IL-4 to IL-4δ2 [41]. In conclusion, whether a person develops into a TB patient after being infected with TB may depend on the expression level of IL-4/IL-4δ2 (Figure 3). Based on the effect of IL-4δ2, it is speculated that it can be used as a new therapeutic approach.

### 4.2. IL-7 and IL-7 R Isoforms Play Different Roles in Control of Mtb Infection

Interleukin-7 (IL-7) and its receptor (IL-7R) play a very important role in the differentiation and growth of immune cells, and they have alternative splicing in infectious diseases [44]. The expression of IL-7 receptor and IL-7 sensitivity of monocytes in the blood of tuberculosis patients are impaired, which has a potential impact on the ability to antimycobacterial [45].

Interleukin-7, which is secreted by macrophages after infection with *Mtb*, stimulates the development of antigen-presenting cells and ensures the persistence of T cells. Non-human primates increased the expression of the IL-7 protein in the lung tissue after the rBCG vaccine [46]. IL-7 can prolong the survival of mice and reduce the *Mtb* load in the lungs [47]. The six exons of IL7 produce nine isoforms by alternative splicing, including IL7δ3, IL7δ4, IL7δ5, IL 7δ3/4, IL 7δ4/5, IL7δ3/4/5, IL7δ3/4-52bp E2, and IL7δ4/5-52bp E2 in addition to full-length IL7c [46,48]. In peripheral blood monocytes of non-human primates, the highest expression level is IL-7c, followed by IL-7δ5 and IL-7δ4 (Figure 3) [46]. IL-7δ5 and IL-7δ4/5 were mainly expressed in the granuloma tissue from patients with latent tuberculosis (Figure 3) [48]. Although the transcription pattern of IL-7 did not change after BCG or virulent *Mtb* infection, the isoforms of IL-7 expression in the granuloma tissue of patients with latent tuberculosis were different. This differential expression suggests that different IL-7 isoforms may be related to the latent state of *Mtb*.

IL-7R has eight exons and generates three isoforms through alternative splicing, namely IL-7Rc, IL-7Rδ6, and IL-7Rδ5/6. The isoforms of IL-7R expression in different tissues of non-human primates are similar, but the proportions are different. The two main expression isoforms are IL-7Rc and IL-7Rδ6. The content of IL-7Rδ6 in the lungs, hilus LNs, and spleen is 90%, 30%, and 10%, respectively [46]. Soluble IL-7R(sIL-7R, IL-7Rδ6) may inhibit IL-7 activity by decreasing phosphorylation signaling [49]. sIL-7R acts as an isoform that inhibits IL-7 activity, and appears to have the opposite effect to full-length IL-7Rc. These opposite isoforms deserve our attention, as it may be the key to the development of TB when infected with *Mtb*. IL-7 reduces lung *Mtb* load in mice, while sIL-7R expression reduces IL-7 activity, allowing uncontrolled reproduction of *Mtb*.

### 4.3. IL12-Rβ1 Is Essential for Host Defence against Mtb Infection

IL-12Rβ1 encodes a transmembrane glycosylated protein and is a low affinity receptor for the IL-12p40 subunit [50]. It is extremely important for humans to resist various intracellular pathogens, including *Mtb* [51]. Human leukocytes express up to 13 different isoforms, of which 2 major isoforms are located on the cell surface and inside the cell, and the other 11 are minor isoforms [52]. In the absence of IL12Rβ1, the cytokines IL-12 and IL-23 could not play a proinflammatory role, and the growth of intracellular bacteria was not inhibited [52].

After infection with *Mtb*, the il12rb1 in dendritic cells produces an IL-12Rβ1ΔTM isoform by alternative splicing [53]. Compared with IL-12Rβ1, IL-12Rβ1ΔTM has altered C-terminal sequences and lacks transmembrane domains, which are localized in the intracellular reticulum similar to the endoplasmic reticulum. IL-12Rβ1ΔTM preserves a signal peptide, cytokine binding domain, and fibronectin domain of IL12Rβ1 [52]. The role of IL-12Rβ1ΔTM is to enhance IL-12Rβ1-dependent dendritic cell (DC) migration and activate *Mtb*-specific T cells [53]. The migration of DCs is required for the activation of naive T cells. Mice lacking IL-12Rβ1ΔTM will have an impact on their ability to control infection with *Mtb* in extrapulmonary lung organs [54]. Increased *Mtb* burden in IL-12Rβ1ΔTM-deficient mice is associated with decreased lymph node cell count and decreased Th1 development [54]. IL-12Rβ1ΔTM enhances resistance to *Mtb* infection by increasing TH1 cells response to IL-12 [54]. It is concluded that the alternative splicing of IL12Rβ1 plays a key role against *Mtb* infection.

### 4.4. The Protective Role for IL-32 Isoforms upon Mtb Infection

Interleukin-32 (IL-32) is a multifunctional cytokine [55], originally described as a transcript (NK4), that is expressed in activated natural killer or T cells [56].The IL-32 gene does not share sequence homology with other cytokines, but is named IL-32 because of its powerful pro-inflammatory effects [57]. The eight exons of IL-32 produce nine isoforms through alternative splicing, namely IL-32 α, IL-32 β, IL-32 γ, IL-32 δ, IL-32 ε, IL-32 ζ, IL-32 η, IL-32 θ, and IL-32s [58]. Among them, the four representative isoforms are IL-32α, IL-32β, IL-32γ, and IL-32δ. These isoforms may have different effects through heterodimeric interactions, but there is no interaction between homologous pairs [58]. There are 10 types of heterodimeric interactions that have been confirmed between IL-32 isoforms, of which IL-32γ is the most common and can interact with IL-32 δ, IL-32 ζ, IL-32 η, and IL-32 θ (Figure 4) [58].

Many studies have shown the role of IL-32 in anti-mycobacterium. *Mtb* induces IL-32 production through a Caspase-1/IL-18/Interferon-γ-dependent mechanism, and the amount produced is positively correlated with the concentration of *Mtb* [59]. IL-32 enhanced the production of proinflammatory cytokines TNF-α, IL-1β and IL-8, and decreased the number of *Mtb* in the infected macrophages [60]. Expression of human IL-32 in mice increases the number of innate and adaptive immune cells, which protects mice from infection by hypervirulent strain of *Mtb* [61]. The expression level of IL-32 in patients with pulmonary tuberculosis was lower than that in healthy people [62]. With the treatment of tuberculosis, the level of IL-32 gradually returned to normal [62]. Compared with healthy individuals, the expression of IL-32 in airway epithelial cells of the lungs infected with *Mycobacterium aviruleum* complex was significantly increased [63]. IL-32γ significantly reduces the growth of *M. avium* in macrophages [63]. The induction of IL-32 expression in monocytes infected with *Mycobacterium leprae* was performed in a NOD2-dependent manner [64]. Our previous research showed that *Mycobacterium smegmatis* with PPE32 significantly increases the expression of IL-32 in macrophages through the ERK1/2 signaling pathway [65].

The defensive role of IL-32 on *Mtb* may be due to the increased apoptosis of macrophages. Caspase-3 is considered to be the most important enzyme in the process of apoptosis. As the activator of Caspase-3, IL-32 is involved in cell apoptosis and reduces the viability of intracellular *Mtb* [60]. The anti-*Mtb* effects of IL-32γ are also mediated through caspase-3-independent apoptosis, including cathepsin- and caspase-1-mediated pyroptosis (Figure 5) [66]. However, the induction of apoptosis is not the only mechanism by which IL-32 prevents *Mtb*. IL-32 induced the vitamin D-dependent antimicrobial peptides cathelicidin and DEFB4 and has antibacterial activity in vitro when 25-hydroxyvitamin D is sufficient (Figure 5) [67]. Cathelicidin and DEFB4 are potent antimicrobial peptides that act against intracellular infection in macrophages [67].

Different IL-32 isoforms have different effects and regulation patterns in anti-tuberculosis treatment. Among these isoforms, IL 32β was the predominantly expressed isoform in activated T cells [68]. IL 32β and IL32γ have been studied extensively [69]. IL-32γ is considered to be the greatest biological active isoform in inducing proinflammatory cytokines, possibly because it contains a full range of exons [70]. Compared with mouse macrophages expressing IL-32γ and IL-32β, macrophages expressing IL-32γ alone were more effective in limiting the growth of *Mtb* [61]. The effect of IL-32 on the prevention of *Mtb* infection may depend on the relative abundance of different IL-32 isoforms [62]. *Mtb* stimulates peripheral blood mononuclear cells (PBMCs) to decrease the expression of IL-32γ and increase the expression of IL-32β [62]. The ratio of IL-32γ/IL-32β correlated positively with IFN-γ and negatively correlated with IL-6, IL-1Ra, and IL-17 [62]. IFN-γ, IL-6, IL-1Ra, and IL-17 have been proven to play an important role in anti-tuberculosis treatment many times. In conclusion, different isoforms of IL-32 have different defense effects against *Mtb*. Considering that the defense effect of IL-32 on *Mtb* may be related to the apoptosis of macrophages, it is worth considering whether different isoforms play different roles by participating in macrophage apoptosis.

## 5. Discussion

In a primary TB infection, not all bacteria are destroyed, even if they are successfully controlled by the immune system. Some bacteria remain dormant in tissues for the rest of the life of the individual. TB latency is actually an active process of protective immune maintenance. This potential state deserves our attention for two reasons. First, in countries with low or moderate TB prevalence, most TB cases are caused by reactivation of latent infections. Second, latent bacteria may be in a physiological state similar to drug-resistant bacteria. So, how to prevent latent infection tuberculosis bacteria from being reactivated is an important question when directing the control of tuberculosis. Keeping *Mtb* in a latent state and preventing the transition to active disease may be an important way to prevent TB. The ratio of IL-4 to IL-4δ2 affects the progression of latent to active tuberculosis [41]. The expression pattern of IL-7 in patients with latent tuberculosis and *Mtb* infection is different [46,48]. Macrophages expressing IL-32γ alone were more effective in limiting the growth of *Mtb* [61]. All these factors suggest that the alternative splicing regulation of immune-related genes may be an important factor affecting the activation or dormancy state of *Mtb*. We speculate that alternative splicing is associated with the progression of latent to active tuberculosis and may be key to whether infected contacts will develop progressive disease or not.

Alteration of host gene expression is not only an effective way for the host to kill mycobacteria, but also a possible strategy for mycobacteria to successfully evade the immune system. An effective and simple way to alter gene expression is through alternative splicing of mRNA. After infection with Mycobacterium tuberculosis, cells undergo changes through 1. directly reducing the expression of splicing factors, 2. altering the original function of proteins through splicing, and 3. expressing unique isoforms [7,32,33,34,35]. These changes enable the bacteria to escape the immune system and exhibit antagonistic effects on immune genes. It may also be intentional by the host to more effectively guide the immune system against *Mtb*. All of these suggest that more study is needed to clarify the exact mechanism of cytokine and its splice variants in anti-TB. This will help to fully understand the immune response to Mycobacterium tuberculosis infection, and is crucial for the development of tuberculosis biomarkers and new drug targets.

## Figures and Tables

**Figure 1 molecules-29-01798-f001:**
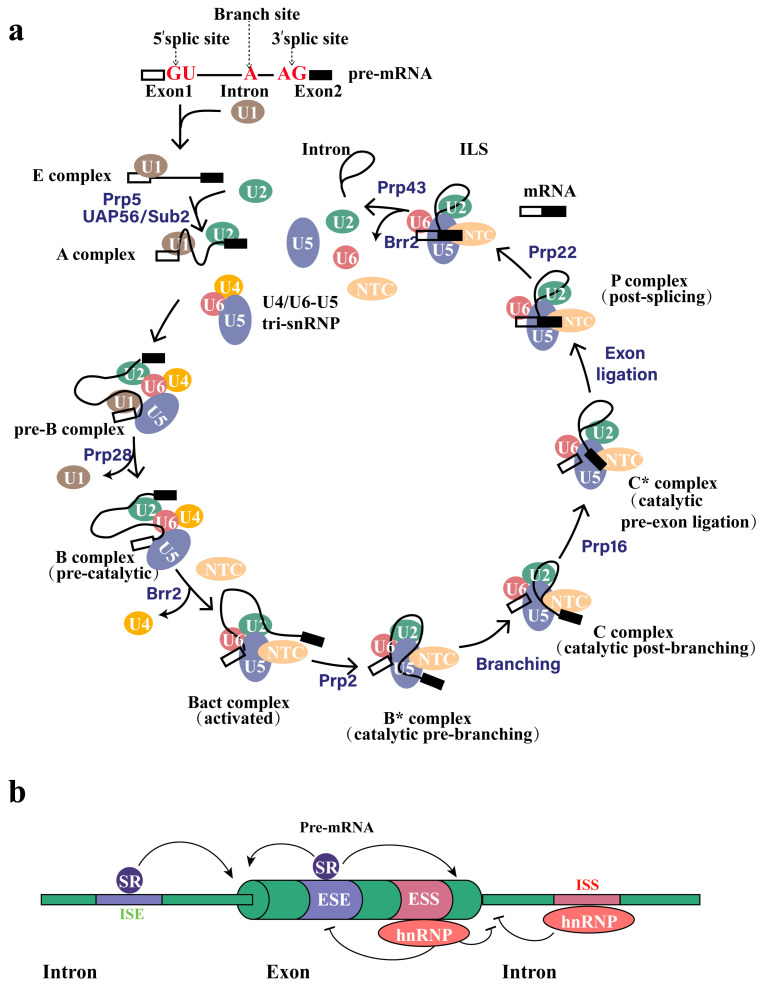
**Overview of splicing reactions and involved regulatory signals/proteins.** (**a**) The assembly and disassembly of the spliceosome during the entire splicing reaction. The figure shows the progressive interaction of spliceosome small ribonucleoprotein (snRNP) particles (U1, U2, U4, U5, and U6; brown, green, yellow, pink, and purple, respectively) during the removal of introns from pre-mRNA containing two exons (white and black). The name of the spliceosome complex and the catalytic steps of the reaction are marked. (**b**) The cis-acting elements stabilize or disrupt the assembly of splicing bodies on the pre-mRNA through positive and negative signals. The diagram illustrates a common segment of eukaryotic precursor mRNA, which comprises one exon flanked by two introns. Factors that promote splicing reactions at nearby splice sites, such as serine-arginine repeat (SR) proteins (shown in dark purple), frequently bind to Intronic and Exonic Splicing Enhancers (ISE and ESE, indicated in purple). Intronic and Exonic Splicing Silencers (ISS and ESS; pink) are often constrained by factors that inhibit splicing from nearby splice sites, such as heterogeneous nuclear ribonucleoprotein particles (hnRNP) proteins (red). The * indicates a catalytically active complex.

**Figure 2 molecules-29-01798-f002:**
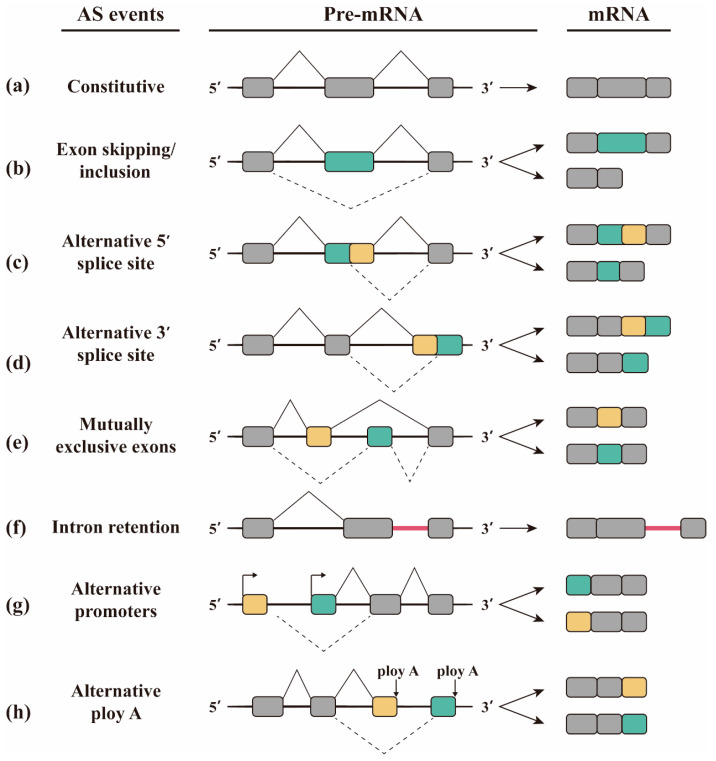
**Graphical description of alternative splicing.** (**a**) Normal splicing. (**b**) Exon skipping or inclusion. (**c**) Alternative 5′SS selection. (**d**) Alternative 3′SS selection. (**e**) Mutually exclusive splicing. (**f**) Retained intron. (**g**) Alternative promoters. (**h**) Alternative polyA. Grey rectangles represent component exons. Yellow and green rectangles represent variable exons. The black solid line indicates introns. The thin solid lines and dotted lines located on exons represent different splicing patterns.

**Figure 3 molecules-29-01798-f003:**
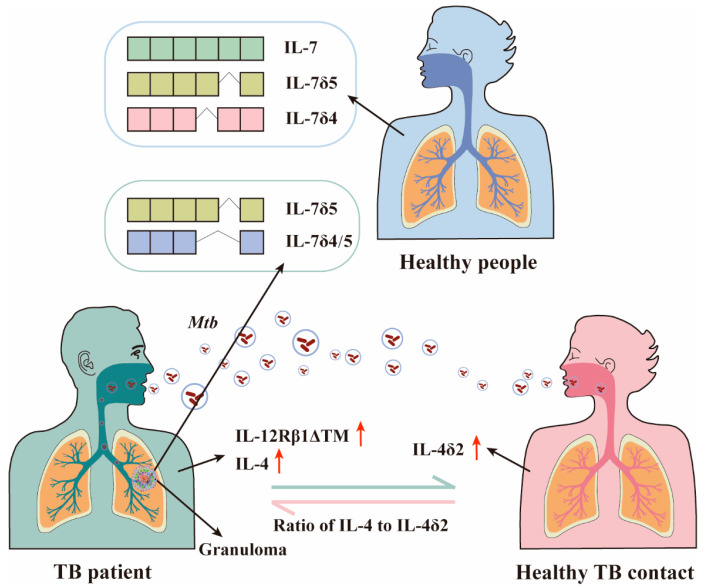
**Alternative splicing of macrophages caused by *Mtb* infection.** The difference in alternative splicing between Healthy people, TB patients, and Healthy TB contacts. The spherical structure of the lungs in TB patients is granuloma. The red arrows indicate increased protein expression.

**Figure 4 molecules-29-01798-f004:**
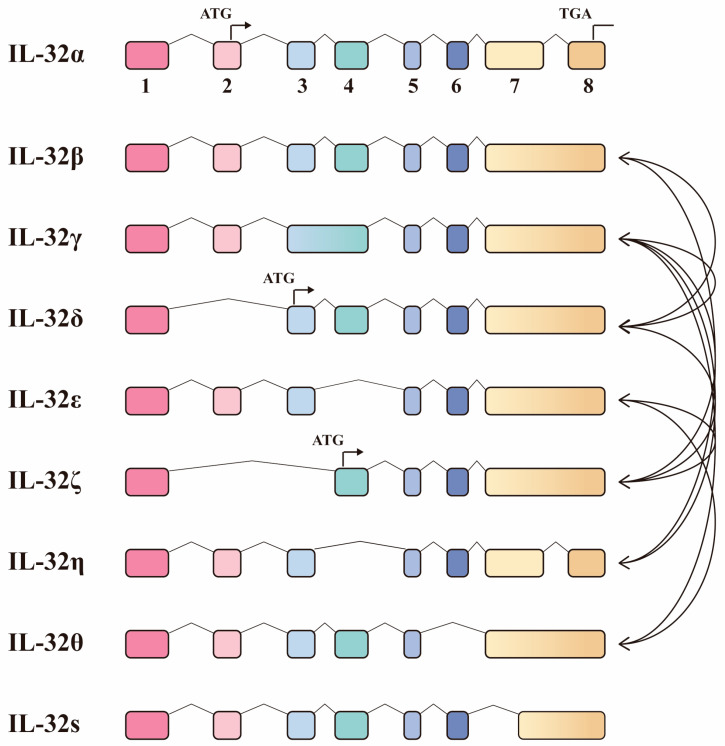
**Different isoforms of IL32 and their interactions.** The nine isoforms of IL-32 are listed from top to bottom: IL-32α, IL-32β, IL-32γ, IL-32δ, IL-32ε, IL-32ζ, IL-32η, IL-32θ, and IL-32s. The 1–8 exons of IL32 are arranged from left to right. ATG is the start codon. TGA is a stop codon. The black arcs with arrows represent interactions between different isoforms.

**Figure 5 molecules-29-01798-f005:**
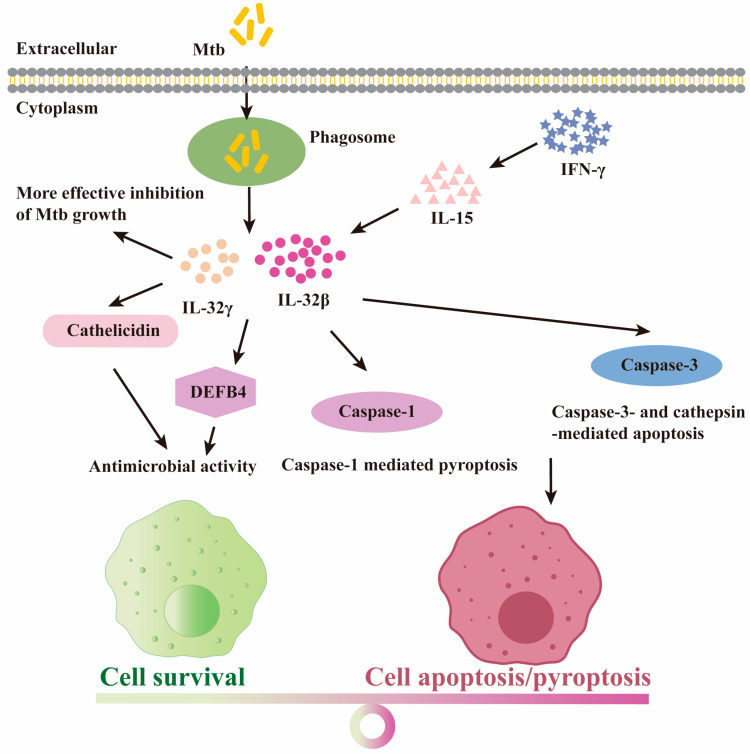
**The protective role for IL-32 isoforms upon *Mtb* infection.** Macrophage response after *Mtb* infection. The yellow rod is *Mycobacterium tuberculosis*. Cathelicidin and DEFB4 are antimicrobial peptides.

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
