# Peer review of "The Role of mRNA Alternative Splicing in Macrophages Infected with Mycobacterium tuberculosis: A Field Needing to Be Discovered"

_molecules, 2024, doi:10.3390/molecules29081798_

Round 1

Reviewer 1 Report

Comments and Suggestions for Authors

This review article “ The role of mRNA alternative splicing in macrophages infected with Mycobacterium tuberculosis: a field need to be discovered” form Hong W et al. has been very well written, they have summarized the currently advanced mRNA splicing through Mycobacterium tuberculosis infection of macrophages, such as directly reducing the expression of splicing factors could effect on regulated gene expression, changed gene truncation or protein domains. The manuscript of this review article will help us understanding the immune response to Mycobacterium tuberculosis infection and they may develop new biomarkers and drug targets. 

Author Response

Dear Reviewer,

      I am deeply grateful for your encouraging feedback on our review article "The role of mRNA alternative splicing in macrophages infected with Mycobacterium tuberculosis: a field to be discovered." Your recognition of our work is truly appreciated.

       Your comments highlight the importance and relevance of our study, which aims to summarize the current understanding of mRNA splicing in macrophages infected with Mycobacterium tuberculosis. We are pleased that you found the manuscript well-written and informative, as we strive to communicate complex scientific concepts in a clear and accessible manner.

       Thank you once again for your valuable feedback. We look forward to continuing our research in this field and making further contributions to the scientific community.

Yours sincerely,

Weiling  Hong

Reviewer 2 Report

Comments and Suggestions for Authors

Hong et al. report that alternative splicing plays a role and has potential importance in Mycobacterium tuberculosis (Mtb)-infected macrophages. Alternative splicing of genes is not only an effective way for the host to eliminate Mtb, but also a possible strategy for mycobacteria to successfully evade the immune system. This report demonstrates that following Mtb infection, cells undergo changes by reducing the expression of splicing factors, altering the original function of proteins through splicing, and expressing specific isoforms. Moreover, alternative splicing regulation of immune-related genes, such as IL-4, IL-7, IL-7R, and Il-12R, may be an important factor affecting the activation or dormancy state of Mtb. In conclusion, Hong et al. discuss the importance of understanding alternative splicing in the human immune response to Mtb for TB control and the development of new targets for vaccines, and drugs. That reports are well-designed, organized, and descripted. But several questions were occurred. I have major questions.

1. Studying alternative splicing in the context of macrophages infected with Mtb is an interesting subject. However, it seems that there may be some hurdles in conducting in vivo experiments due to the differences in patterns observed between humans and mice. I am curious about the potential challenges associated with using alternative splicing to study Mtb infection.

2. I am curious about the advantages of developing biomarkers using alternative splicing, especially in terms of diagnosis compared to traditional biomarkers for Mtb infection.

3. The existing figures alone do not make it easy to identify which alternative splicing isoforms are upregulated or downregulated in Mtb-infected macrophages simultaneously. It would be more beneficial to add a table summarizing the information related to the alternative splicing isoforms mentioned in the report.

4. It seems that there is a need for further description of the mechanism by which MTB infection regulates alternative splicing of genes. It would be beneficial to include additional information regarding Mtb-related factors or mechanisms associated with MTB-mediated regulation of alternative splicing in macrophages identified to date.

5. Recently, there has been a growing interest and concern about multidrug-resistant tuberculosis (MDR TB) and nontuberculous mycobacteria (NTM) infections. I am curious about the similarities and differences between alternative splicing associated with Mtb infection and other conditions.

Comments on the Quality of English Language

Round 2

Reviewer 2 Report

Comments and Suggestions for Authors

My concerns has been addressed